# Effects of Tetracycline and Copper on Water Spinach Growth and Soil Bacterial Community

Jiadan Tao [1,2], Jiayu Wang [1,2], Xiongkai Zheng [1,3], Aiping Jia [1,2,*], Mengyao Zou [1,2], Jinlian Zhang [3,4] and Xueqin Tao [1,2,*]

1.  College of Resources and Environment, Zhongkai University of Agriculture and Engineering, Guangzhou 510225, China; tjd18971073258@163.com (J.T.); wjy2473745004@163.com (J.W.); zheng_xiongkai@163.com (X.Z.); mengyaozou_zhku@126.com (M.Z.)
2.  Engineering and Technology Research Center for Agricultural Land Pollution Prevention and Control of Guangdong Higher Education Institutes, Zhongkai University of Agriculture and Engineering, Guangzhou 510225, China
3.  School of Environment and Energy, South China University of Technology, Guangzhou 510006, China; jinlianzhang@scut.edu.cn
4.  The Key Lab of Pollution Control and Ecosystem Restoration in Industry Clusters, Ministry of Education, South China University of Technology, Guangzhou 510006, China
*   Correspondence: jap7806x@126.com (A.J.); taoxueqin@zhku.edu.cn (X.T.)

**Abstract:** The effects of tetracycline (TC) and copper (Cu) on the growth of water spinach and the bacterial community structure in soil were examined in this study. The results revealed that a single Cu treatment decreased water spinach development more severely than TC, and that the toxic effects of TC and Cu on water spinach were synergistic at low doses and antagonistic at high concentrations. The single Cu treatment had the largest influence on the activities of three antioxidant enzymes (Superoxide Dismutase (SOD), Peroxidase (POD), Catalase (CAT)) and the content of Malondialdehyde (MDA) in water spinach leaves, followed by the TC and Cu composed treatment, with the single TC treatment having the least effect. The results of 16Sr RNA sequence analysis showed that the richness and diversity of soil bacterial communities were reduced by either a single TC or Cu treatment. Cu had a greater effect on the composition of the microbial community at genus level than TC. In conclusion, Cu had a greater influence on the growth of water spinach and soil microbial community composition than TC. TC and Cu exhibited synergistic effects at low concentrations and antagonistic effects at high concentrations on relevant indicators when Cu concentration was fixed.

**Keywords:** tetracycline; copper; water spinach; antioxidant enzymes; microbial community



## 1. Introduction

Antibiotics are widely used in livestock farming and the aquaculture industry because of their antibacterial and bactericidal effects [1,2]. Tetracycline constitutes a family of broad-spectrum bacteriostatic antibiotics that are widely used in livestock and poultry breeding and exist in large quantities in the environment [3]. China is one of the largest antibiotics-consuming countries, producing about 210,000 tons of antibiotics annually, nearly half of which are used in livestock and poultry breeding [4,5]. Many antibiotics are poorly absorbed in the gut, cannot be metabolized completely, and most of them are excreted as parent compounds or metabolites into soil and water environment [6]. In the soil environment, a portion of antibiotics will be degraded as microbial carbon sources, while the other portion will be absorbed into the soil, causing toxicity to surrounding plants and animals [7–9]. Zhou et al. [10] investigated antibiotic residue in animal manure collected in and around downtown Nanjing, China, and found that tetracycline (TC) was present in most of the samples with a concentration of up to 1920 mg kg$^{-1}$. TC inhibits

protein synthesis in bacteria by binding to the 30S subunit A site of bacterial ribosomes in the environment, and it also interferes with the function of ribosomes in agricultural crops mitochondria and chloroplast organelles, thus inhibiting bacterial growth and reproduction and affecting agricultural crops [11,12].

As an animal feed additive and agricultural fertilizer, Cu can also cause pollution of heavy metals in the environment due to excessive practical application [13,14]. Li et al. found that Cu concentration in about 21.02% of sampling sites exceeded the screening value of China's latest national standard (50 mg kg$^{-1}$) by analyzing 1731 investigation sites in Chinese farmland [15]. In China, copper in animal manure accounted for 69% of the total input of different agricultural soil pollution sources [16]. Cu is an essential metal for the normal growth and development of plants, but excessive Cu can be toxic to plants [17]. In the process of plant growth, Cu can affect plant growth processes by interfering with SH groups or related enzymes involved in catalysis [18]. The toxic mechanism of Cu to microorganisms was reflected in the combination of Cu and microbial enzyme molecules, which would not only inhibit the growth and metabolism of microorganisms but also lead to the change of microbial community structure [19].

Animal manure is commonly used as an organic fertilizer input to soil to improve the nutrient properties of soils because it is rich in nitrogen, phosphorus and organic matter [20,21], which directly contributes to the accumulation of antibiotics and heavy metals in soils [22–24]. In recent years, the complex interaction of antibiotics and heavy metals has gained increasing attention [25,26]. Studies have shown that the antagonistic effect of enrofloxacin and Cu-composed treatment was higher than the synergistic effect on soil microorganisms [27]. Combined sulfadoxine and Cu contamination exhibited higher toxicity and reduced the biomass of soil bacteria and actinomycetes [28]. After 20 days, the combined effect of TC (15 mg L$^{-1}$) and Cu (1 mg L$^{-1}$) showed a synergistic effect on the biomass of water hyacinth [29]. The combined contamination with low concentrations of TC (10 μmol L$^{-1}$) and Cd (5 μmol L$^{-1}$) promoted the growth of rice roots and showed an antagonistic effect [30]. The studies referenced above showed that the ecological effects of antibiotics and heavy metals on different crops and soil microorganisms when coexisting are still controversial. Therefore, TC and Cu were selected as targeted antibiotics and heavy metals in this study. Water spinach (*Ipomoea aquatica Forskwas*) was used as a test plant. Specifically, we investigated the following: (1) the impact of TC and Cu on the growth and physiological characteristics of water spinach through a pot experiment; (2) the effect of TC and Cu on soil bacterial community; (3) the relationship between TC and Cu pollution factors and microbial community structure. These findings will obtain useful information about the harm of TC and Cu-combined pollution to crops and soil micro-ecology, and they will provide scientific basis for solving the problem of combined soil antibiotics and heavy metals pollution.

## 2. Materials and Methods

### 2.1. Experimental Material

Water spinach seeds were purchased from Beijing Dongsheng Seed Co., Ltd. Initially, the seeds were soaked in a constant temperature water bath at 50 °C for 15–20 min, sterilized by using solution of H$_2$O$_2$ at 10% (*w/v*) for 15 min, and slightly rinsed with distilled water. Seeds immersed in deionized water at 30 °C until dewy.

Test soil was collected from the surface (0–20 cm) of a vegetable field in Panyu, Guangzhou, Guangdong, China. After they were air dried, the soil samples were passed through a 2 mm nylon sieve, homogenized, and used for potting experiments. The physico-chemical properties of the test soils are shown in Table 1. TC was added by spraying with ultrapure water, and Cu was added by spraying in the form of copper sulfate solution; each parallel group was treated and mixed thoroughly, and the soil was placed in a place protected from light and ventilated for 24 h after poisoning.

**Table 1.** Physical and chemical properties of test soils.

| Determination Index | Measured Value |
|---|---|
| pH | 5.44 |
| SOM (g kg$^{-1}$) | 26.18 |
| TC (mg kg$^{-1}$) | Not Detected |
| Cu (mg kg$^{-1}$) | 35.00 |
| Total nitrogen (g kg$^{-1}$) | 0.84 |
| Available N (A-N, mg kg$^{-1}$) | 181.81 |
| Available P (A-P, mg kg$^{-1}$) | 87.16 |
| Available K (A-K, mg kg$^{-1}$) | 514.27 |

*2.2. Experimental Design and Treatment Plan*

Pot experiments were set up using plastic pots with length 16 cm, width 16 cm, and height 14 cm, each with a nylon screen (100 mesh) at the bottom. Soil (2.0 kg) was added to each pot, and six seeds were sown per pot, then placed in a ventilated and cool place for a week. They were then moved to the greenhouse, and three plants per pot were set up; water was added regularly to keep the soil moisture at about 60% of the field water holding capacity during the incubation period. The greenhouse temperature was 35 °C during the day and 25 °C at night. Plants were harvested after 30 days of planting. The experimental TC and Cu concentrations were set up based on the TC and Cu concentrations when the pre-experimental root length inhibition rate reached 10–60% (The results are shown in Table S1), and the design of the treatments are shown in Table 2. Each treatment was set up with three replications, and the treatments without TC and Cu were used as control.

**Table 2.** Design of experimental treatment groups.

| TC (mg per kg of Soil) | Cu (mg per kg of Soil) | | |
|---|---|---|---|
| | 0 | 100 | 300 |
| 0 | Control | Cu1 | Cu3 |
| 100 | TC1 | TC1-Cu1 | TC1-Cu3 |
| 300 | TC3 | TC3-Cu1 | TC3-Cu3 |
| 500 | TC5 | TC5-Cu1 | TC5-Cu3 |

*2.3. Analytic Methods*

2.3.1. Assay of Physiological and Biochemical Indexes

The activity of antioxidant enzymes, as the main substances that maintain the balance of reactive oxygen species in water spinach, can reflect the degree of stress in water spinach in response to environmental stresses [31]. MDA is an important byproduct of membrane lipid peroxidation. MDA content can indicate the extent of damage to plant cell membranes [32].

The height of water spinach plants was measured every 7 days during the 30-day incubation period. After harvesting, water spinach was washed under running water and blotted out with absorbent paper before being divided into above-ground and below-ground parts and weighed. After harvesting, the activity of Superoxide Dismutase (SOD), Peroxidase (POD), Catalase (CAT) and the content of Malondialdehyde (MDA) were measured in the leaves of water spinach using kits purchased from Beijing Ltd.

2.3.2. Soil Physical and Chemical Indicators

Soil properties, such as pH, SOC, available K (A-K), available P (A-P) were determined with the method described by Lu [33]. Total Cu in soil was quantified by reference to the Chinese environmental protection standard-total metal elements in soil and sediment microwave digestion method (HJ 832-2017), using HCl-HNO$_3$-HF digestion and a flame-type atomic absorption spectrophotometer (TAS-990, PERSEE). National standard material

of soil (GBW-070009) and blank test were used for quality control of the sample analysis process, and the recovery of Cu was 90.1%~105.3%.

TC in the soil were extracted using a solid-phase extraction column (CNW HLB; 200 mg, 6 mL) according to the methods developed by Tang et al. [34]. The aqueous concentrations of TC were determined using reversed-phase high-performance liquid chromatography (G7111A, Aglient) and UV-visible detection at the wavelength of 350 nm, with a symmetry C18 column ($4.6 \times 150$ mm, 5 μm). The injection volume was 10 μL and the mobile phase comprised a mixture of methanol and water containing 0.01 mol L$^{-1}$ oxalic acid, which was maintained at a flow rate of 1 mL min$^{-1}$, and the recovery of TC was 80.6%~102.3%.

### 2.4. 16S rRNA Gene Analysis via Illumina High-Throughput Sequencing and Data Analysis

Total genomic DNA samples was extracted using the EZNA Gel Extraction Kit (Omega Bio-Tek, Norcross, GA, USA), following the manufacturer's instructions, and stored at $-20$ °C prior to further analysis. The quantity and quality of extracted DNAs were measured using a NanoDrop NC2000 spectrophotometer and agarose gel electrophoresis, respectively.

The universal bacterial primers used in this study for microbial community diversity analysis (16S rRNA) were 515F (GTGCCAGCMGCCGCGGTAA) and 806R (GGAC-TACHVGGGTWTCTAAT) for PCR amplification. The PCR components contained 5 μL of buffer ($5\times$), 0.25 μL of Fast pfu DNA Polymerase (5 U/μL), 2 μL (2.5 mM) of dNTPs, 1 μL (10 uM) of each Forward and Reverse primer, 1 μL of DNA Template, and 14.75 μL of ddH$_2$O. The reaction conditions were set as follows: 94 °C predenaturation for 5 min; 94 °C denaturation for 30 s, 25 cycles; 52 °C annealing for 30 s; 72 °C extension for 30 s, 25 cycles; and 72 °C extension for 10 min. Sample-specific 7-bp barcodes were incorporated into the primers for multiplex sequencing.

The resulting samples were subjected to paired-end sequencing on the Illumina Hiseq 2500 platform. At the 97% similarity level, the sequences were grouped into multiple OTUs based on Vsearch clustering, and the OTUs with 97% similarity were selected to generate the expected sparse curves (Rarefaction curve). Based on the results of OTU clustering analysis and using the software QIIME2 (2019.4), Alpha diversity analysis was performed to calculate the richness index Chao1, the observed species, and the diversity index Simpson and Shannon indices, respectively. In addition, the silva_132 database was selected for comparison, and the Bayesian inference algorithm was used to perform taxonomic analysis of species at each taxonomic level for the OTU representative sequences, to count the community composition of each sample, and to perform statistical analysis of the community structure of different samples at each taxonomic level.

### 2.5. Statistical Analysis

The information in the graphs and tables was presented as three triplicate averages ±standard deviations of each treatment. Significant differences among different treatments were performed by SPSS 20, according to the Duncan test ($p < 0.05$). Microbial-community-level data and environmental factor data were uploaded to the QIIME2 online analytic platform for redundancy analysis (RDA) to investigate the association between soil bacterial communities and environmental factors.

## 3. Results and Discussion

### 3.1. Effects of TC and Cu Complex Contamination on Physiological and Biochemical Indicators of Water Spinach

#### 3.1.1. Plant Height and Biomass of Water Spinach

The physiological state of water spinach is an intuitive presentation of its adaptive growth in the environment, which can reflect the impact of pollution stress on plants. The effects of TC and Cu combined pollution on the biomass and plant height of water spinach are shown in Figure 1. It can be seen from Figure 1 that the plant height and

biomass of water spinach were inhibited to varying degrees with the increase of TC and Cu pollution concentrations. According to the comparison of data between groups, the inhibition of single Cu pollution on the growth of water spinach was more obvious than that of TC pollution. When the Cu concentration was 300 mg kg$^{-1}$ the plant height and growth of water spinach decreased by 82.13% and 89.19%, respectively. Compared with the control, single TC stress had no significant effect on plant height and growth of water spinach. When the concentration of Cu was 100 mg kg$^{-1}$, the height of water spinach plants in the TC1-Cu1, TC3-Cu1 and TC5-Cu1 treatment groups was significantly increased by 17.75%, 16.36% and 7.34%, while the growth of water spinach in the corresponding treatment groups was increased by 89.57%, 91.64% and 88.13%, respectively. This suggests that TC could alleviate the stress of Cu on the growth of water spinach and that there might be an antagonistic effect of TC and Cu on the growth of water spinach. The data of TC1-Cu3, TC3-Cu3 and TC5-Cu3 treatments showed that the height of water spinach was decreased by 83.93%, 82.13% and 79.42%, respectively, and the biomass of water spinach was decreased by 89.57%, 91.64% and 88.13%, respectively, as compared with the control. Indicating the high concentration of compound pollution had a significant inhibitory effect on the growth of water spinach.

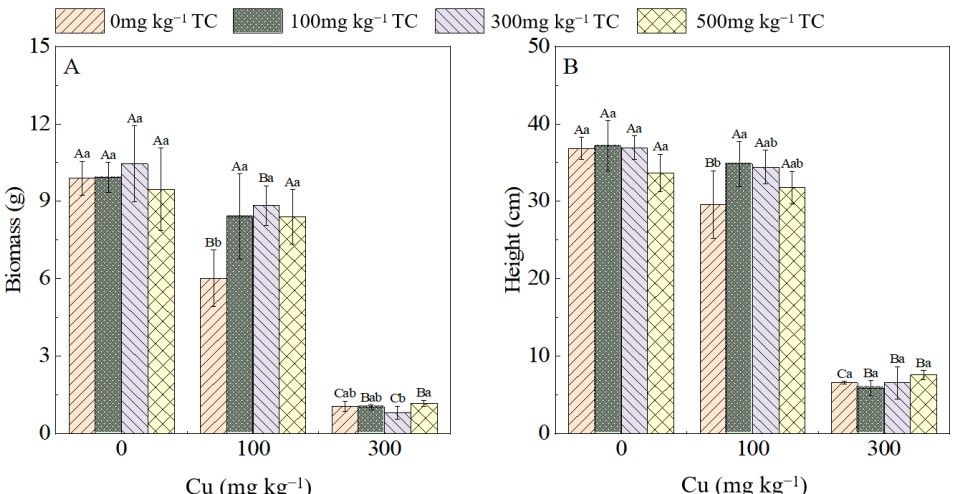

**Figure 1.** Effect of TC and Cu complex pollution on height and biomass of water spinach. (**A**) biomass; (**B**) height. (Different capital letters represents significant differences between different Cu treatment at *p* < 0.05, while different lowercase letters represents TC's.)

### 3.1.2. Antioxidant Enzyme Activity and MDA Content in Leaves of the Water Spinach

To explore the detoxification response of water spinach in response to TC and Cu combined contamination, the activity of related antioxidant enzyme and the content of MDA in the leaves of water spinach were quantified (Figure 2). The activity of POD increased and then decreased under a single TC treatment, while the activity of POD showed a gradual increase with a single Cu treatment. Furthermore, among the composite pollution treatment groups, the activity of POD in all groups except the TC5-Cu3 group showed an increasing trend with the increasing concentration of TC. The results clearly revealed that the complex contamination could improve the activity of POD in water spinach. It also showed that there is a certain threshold for the synergistic or antagonistic effect of TC and Cu on the activity of POD in water spinach.

Generally, water spinach could regulate CAT activity to resist the stressful effects of environmental pollutants. The activity of CAT in water spinach showed a trend of increasing and then decreasing under either single TC or Cu treatment (Figure 2B). When the concentration of Cu in the soil was 300 mg kg$^{-1}$, the activity of CAT in the groups of compound pollution treatment showed a significant decreasing with the increasing concentration of TC, with the lowest activity of CAT in TC5-Cu3 treatment. The same trend

was observed at the concentration of 100 mg kg$^{-1}$ and below of Cu in soil, implying that TC and Cu showed synergistic effects at low concentrations and antagonistic effects at high concentrations on the toxicity of water spinach. The trend of SOD activity in Figure 2C is similar to that in Figure 2B, which also supports the above conclusion. Gao [35] also found that the toxicity of TC and Pb, sulfadiazine and Cu to microcystis aeruginosa would show synergism effects at low concentrations and antagonism effects at high concentrations. For the content of MDA, both single TC or Cu treatments showed an increasing trend (Figure 2D), indicating that both single TC or Cu treatments increased the toxicity on water spinach. The content of MDA decreased with increasing addition of TC when the concentration of Cu was 300 mg kg$^{-1}$, which further demonstrated that TC and Cu compound pollution had some antagonistic effects on the growth of water spinach.

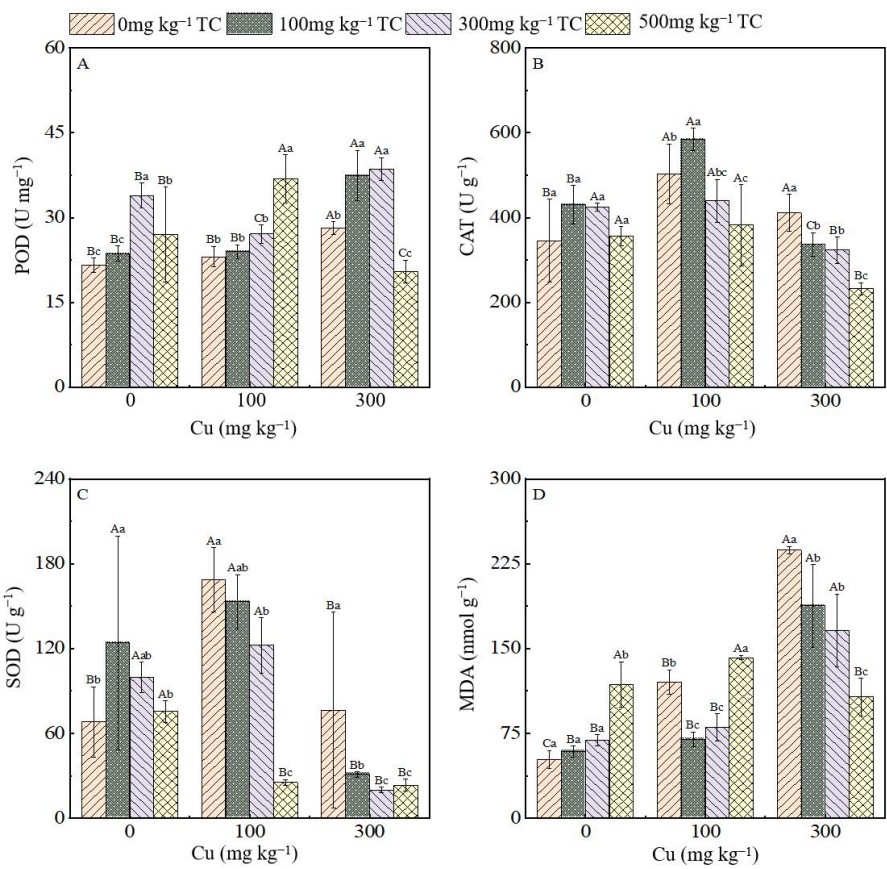

**Figure 2.** Impact of TC and Cu pollution on antioxidant enzyme activity and MDA of water spinach. (**A**) POD; (**B**) CAT; (**C**) SOD; (**D**) MDA. (Different capital letters indicate significant differences between different Cu treatment at $p < 0.05$, while different lowercase letters indicate different TC's.)

The above results indicated that TC and Cu-combined pollution had a stimulating effect on both the activity of antioxidant enzyme and the content of MDA in water spinach. Two-way ANOVA analysis revealed that both TC and Cu had an effect on activity of antioxidant enzyme and content of MDA in water spinach (Table 3). On the whole, the main factor affecting the antioxidant enzyme activity of water spinach was Cu, followed by TC-Cu interaction, with TC having the smallest effect. The results of the ANOVA on MDA were consistent with the above. It indicated that the stress effect of single Cu on water spinach was higher than that of single TC and complex contamination.

**Table 3.** Two-way ANOVA of TC and Cu on antioxidant enzyme activity and MDA in water spinach.

| Factor | TC | | Cu | | TC × Cu | |
|---|---|---|---|---|---|---|
| | F | P | F | P | F | P |
| CAT | 7.988 | 0.001 * | 24.045 | 0.000 * | 11.443 | 0.000 * |
| POD | 24.433 | 0.000 * | 32.009 | 0.000 * | 21.053 | 0.000 * |
| SOD | 1.079 | 0.377 | 18.476 | 0.000 * | 7.370 | 0.000 * |
| MDA | 9.378 | 0.000 * | 56.168 | 0.000 * | 41.781 | 0.000 * |

* Represents significance, $p < 0.05$.

### 3.2. Effect of TC and Cu Composed Pollution on Soil Bacterial Community

### 3.2.1. Alpha Diversity of Soil Bacterial Communities

A total of 237 phyla, 627 orders, 1322 orders, 1904 families, 2341 genera and 416 species were observed by sequencing analysis in 36 soil samples, meeting the requirements for analysis such as Alpha diversity and community structure composition in this research. The indexes of alpha diversity of soil bacterial communities under TC and Cu composed contamination are presented in Figure 3.

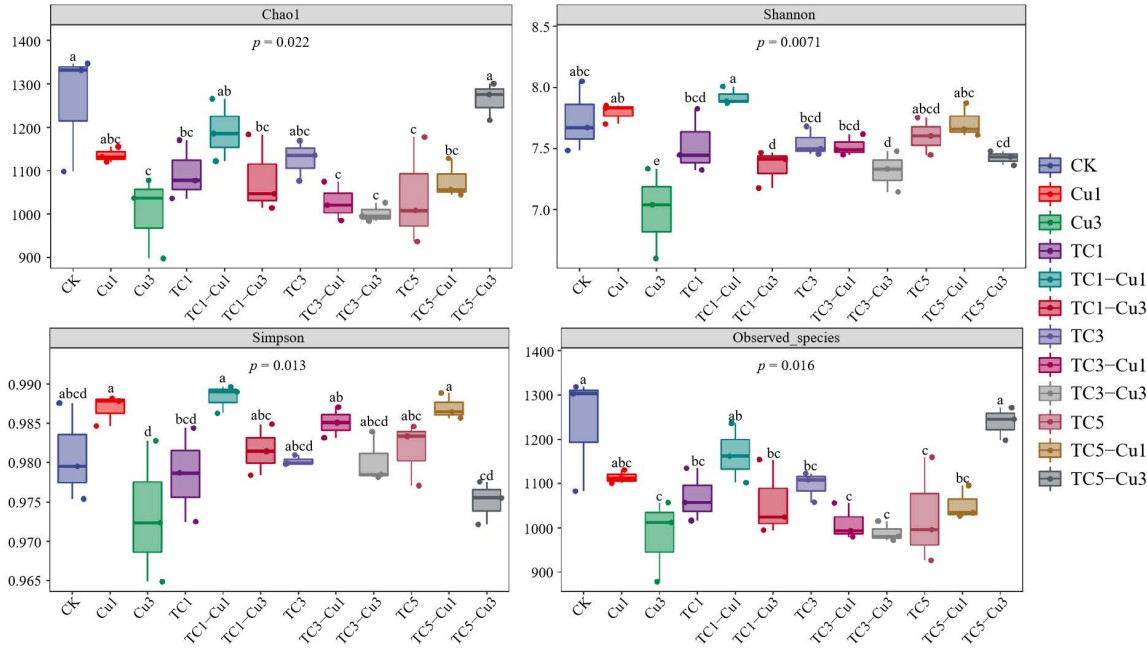

**Figure 3.** Box-line plot of alpha diversity of soil bacterial communities under TC and Cu contamination. Different letters indicate significant difference between different treatments ($p < 0.05$).

In the single Cu treatments, Chao1, Observed species, Shannon and Simpson indexes showed a decreasing trend with increasing concentrations of Cu, indicating that high concentration of Cu resulted in a significant decrease in soil bacterial community richness. The same can be shown for the decrease in richness of bacterial community in soil due to the addition of high concentration of TC. Cao et al. [36] found that Cu at a concentration of 450 mg kg$^{-1}$ significantly reduced the richness and diversity of bacterial communities. Zheng et al. [37] found a similar effect of TC on soil microbial communities. It was consistent with the findings of this research. In the composed treatment groups, when the TC concentration was 100 mg kg$^{-1}$, the indexes of Chao1, Observed species, Shannon and Simpson showed a trend of increasing and then decreasing with the increasing concentration of Cu, which was consistent with the trend of growth of water spinach. Furthermore, as a whole, except for the TC5-Cu3 group, the Observed species and Chao1 indices of the remaining groups showed different degrees of decrease, which indicated that the species richness of the TC5-Cu3 treatment group did not differ much from that of the control group,

and to some extent, it indicated the antagonistic effect of TC and Cu at high concentrations. In addition, there was a significant decrease in Observed species and Chao1 indexes in all treatment groups except the TC5-Cu3 group, which not only indicated that the microbial richness of the TC5-Cu3 treatment group was not significantly different from the control group but also reinforced the antagonistic effect of TC and Cu at high concentrations.

### 3.2.2. Taxonomic Composition of Soil Bacterial Communities

Relative abundance map of soil bacterial communities at the phylum level (Figure 4) showed that there were nine dominant phyla accounting for more than 1%, namely *Proteobacteria* (26.49–39.80%), *Patescibacteria* (6.84–23.8%), *Chloroflexi* (13.67–20.70%), *Actinobacteria* (14.7–17.42%), *Acidobacteria* (5.03–10.76%), *Gemmatimonadetes* (2.94–10.88%), *Planctomycetes* (1.17–2.09%), *Firmicutes* (1.15–2.76%) and *Bacteroides*(0.69–2.14%). In the single Cu treatments, the relative abundance of *Patescibacteria* and *Nitrospirae* decreased and that of *Acidobacteria* and *Gemmatimonadetes* increased with increasing concentration of Cu. Additionally, the relative abundance of *Proteobacteria* in the Cu3 treatment increased by 12.93% compared with the control group. *Proteobacteria* and *Acidobacteria* have been reported as phylum of the heavy metal tolerant bacteria [38,39]. In the single TC treatments, the percentage of *Patescibacteria* and *Bacteroidetes* decreased as the TC concentration increased, and the percentage of *Actinobacteria* in the TC5 treatment increased by 1.17% compared to the control. In the complex treatments, *Bacteroidetes* in the TC1-Cu1 treatment increased by 1.07% compared to the control, and the relative abundance of *Bacteroidetes* was the highest among all groups. *Actinobacteria* and *Gemmatimonadetes* in TC5-Cu3 treatment increased by 2.01% and 5.53%, respectively, compared to the control. The relative abundance of *Actinobacteria* and *Gemmatimonadetes* increased by 2.01% and 5.53%, respectively, and the relative abundance of the two bacteria was the highest among all treatment groups.

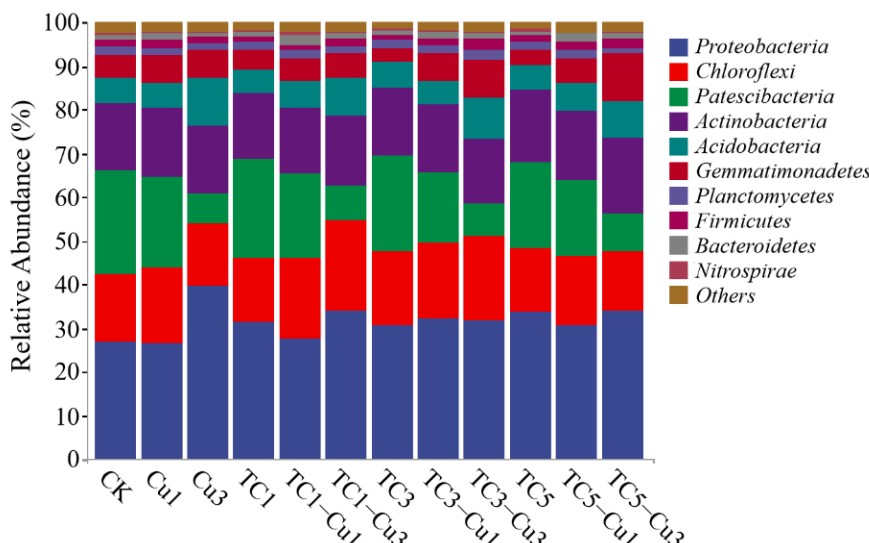

**Figure 4.** Relative abundance of soil bacterial communities at the phylum level under TC and Cu contamination.

Relative abundance map of soil bacterial communities at the genus level (Figure 5) revealed that the observed dominant genus mainly included *Saccharimonadales, JG30-KF-AS9, Mizugakiibacter, Chujaibacter, Gemmatirosa, Acidipila, Halomonas, Sphingomonas, Dyella* and *Acidibacter*. Reddy et al. [40] considered *Acidibacter* as the dominant genus at the genus level in Ganges water, in India, contaminated with antibiotics and heavy metals, which were consistent with the findings of the present research. In the single TC treatments, the relative abundance of *JG30-KF-AS9, Acidibacter* and *Mizugakiibacter* was higher in TC1, TC3 and TC5 treatment groups than in the control soil, and the relative abundance of *Chujaibacter* and *Gemmatirosa* were both higher in the single Cu treatments than in the

control group, and the relative abundance of *Mizugakiibacter* was lower than the control. This might be due to the fact that *Chujaibacter* and related *Metallibacterium scheffleri* belong to the genus of metal metabolizing bacteria [41,42]. The relative abundance of *Mizugakiibacter* and *Saccharimonadales* was lower and that of *Dyella* was higher in the complex treatments compared to the control group. The relative abundance of *JG30-KF-AS9* was higher than that of the single TC or Cu treatments and higher than that of the control group.

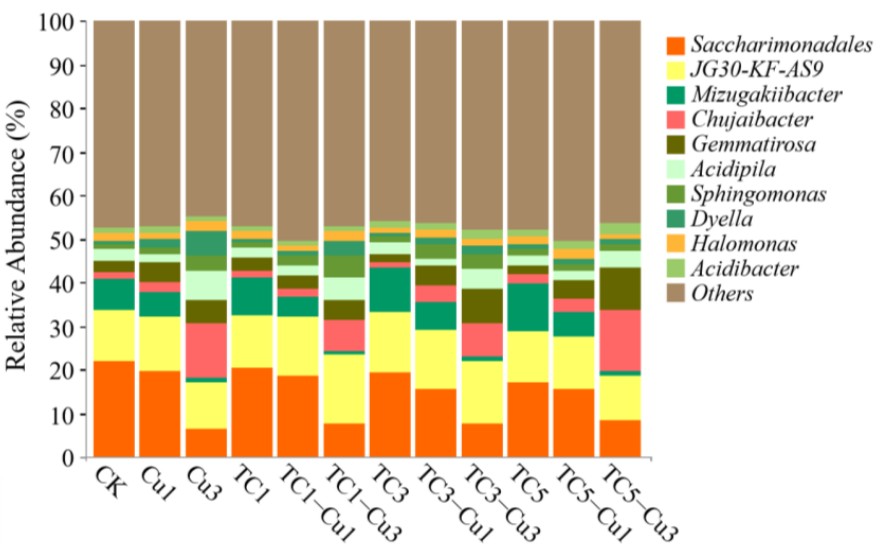

**Figure 5.** Relative abundance of soil bacterial communities at the genus level under TC and Cu contamination.

3.2.3. Clustering and Correlation Analysis between Treatment Groups and Soil Microbial Communities at the Genus Level

The clustering heat maps of microbial at genus level with treatment groups are illustrated in Figure 6. TC3, Control and TC1 treatments as one cluster, TC5 and TC5-Cu1 as another cluster, which indicated that the effect of TC on the soil microbial structure at genus is negligible up to 300 mg kg$^{-1}$, but TC at high concentrations (500 mg kg$^{-1}$) could change the microbial communities. As shown by the clustering of TC3-Cu1, TC1-Cu1 and Cu1 treatments into one cluster and TC3-Cu3, TC1-Cu3 and Cu3 treatments into another cluster, the composition of bacterial community at the genus level at the same concentration of Cu was similar. Indicating that the composition of bacterial community at the genus was dominated by Cu. The community structure of soil bacteria at the genus level is hardly altered by TC at the experimental concentration level. This is consistent with the fact mentioned above that TC exerts a synergistic effect on water spinach at low concentrations and an antagonistic effect at high concentrations under the fixed concentration of Cu. The TC5-Cu3 group, as a single cluster, had more dominant genera showing strong correlation compared to the Cu3-related treatments, such as *Acidothermus*, *Streptomyces*, *Acidibacter*, *Chujaibacter*, *Gemmatirosa*, and *Subgroup_6 genera*. In addition, TC5 and TC5-Cu1 treatment were clustered together, further confirming that high concentrations of TC could alter the composition of bacterial community at genus level. In summary, Cu had a greater impact on the composition of microbial community than TC, and TC had a negligible effect on community composition at the genus level at low concentrations and a greater effect at high concentrations.

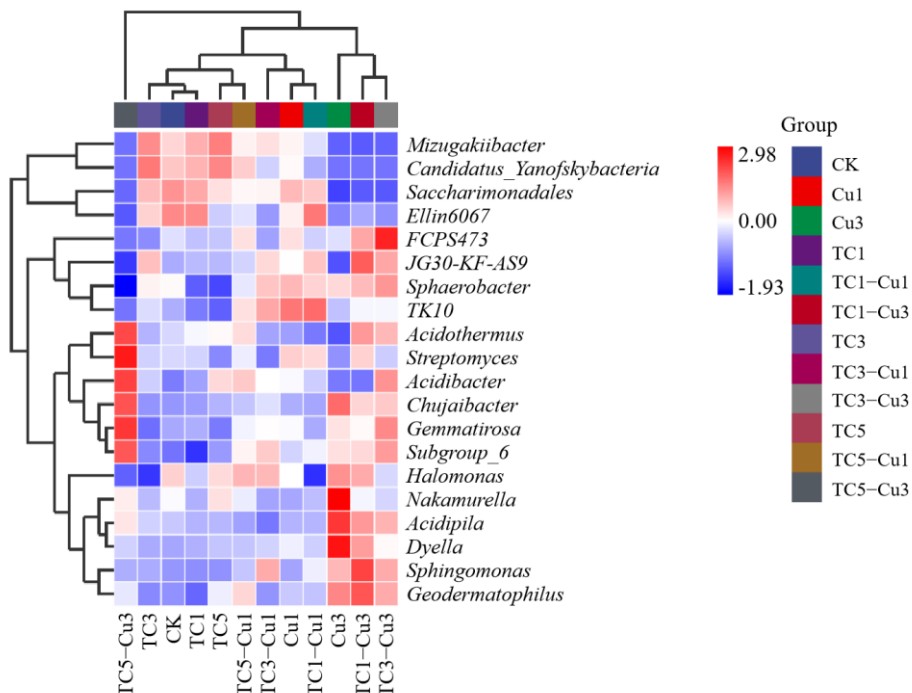

**Figure 6.** Clustering heat map analysis of the correlation between microbial community (genus) and treatments.

3.2.4. Redundancy Analysis

The redundancy analysis (RDA) of bacteria community structure at genus level and environmental factors is shown in Figure 7. Both TC and Cu could change the structure of bacteria community. Cu had the greatest impact on the TC5-Cu3 and TC1-Cu3 treatments; TC had the greatest effect on the TC3-Cu1 and TC5-Cu1 treatments. Furthermore, the genera with strong positive correlations with TC were *JG30-KF-AS9* and *Acidibacter*; the genera with strong negative correlation were *Halomonas* and *Dyella*. The genera with strong positive correlation with Cu were *Chujaibacter* and *Gemmatirosa*, and the genera with strong negative correlation were *Saccharimonadales* and *Mizugakiibacter*.

*Saccharimonadales* were associated with nitrogen cycling and were the dominant genus at higher phosphorus concentrations [43,44]. In this research, abundance of *Saccharimonadales* was positively correlated with biomass of water spinach and negatively correlated with the concentration of Cu. When the concentration of Cu was 300 mg kg$^{-1}$, the relative abundance of *Saccharimonadales* and the biomass of water spinach decreased significantly, which indicated that *Saccharimonadales* could not adapt to the stress of high concentration of Cu. The relative abundance of *Mizugakiibacter* in single Cu treatment in this research was lower than control soil, which is consistent with the previous finding. Xie et al. [45] found that the relative abundance of *Mizugakiibacter* in soil under high Cd stress was lower than the control. It was still unclear about the ecological functions of *Mizugakiibacter*, but some studies have found that *Mizugakiibacter* could reduce the occurrence of plant diseases [46]. *JG30-KF-AS9* belongs to the Ktedonobacteria class, which is anaerobic and antibiotic-resistant and may have some degradation effect on organic pollutants in soil [47].

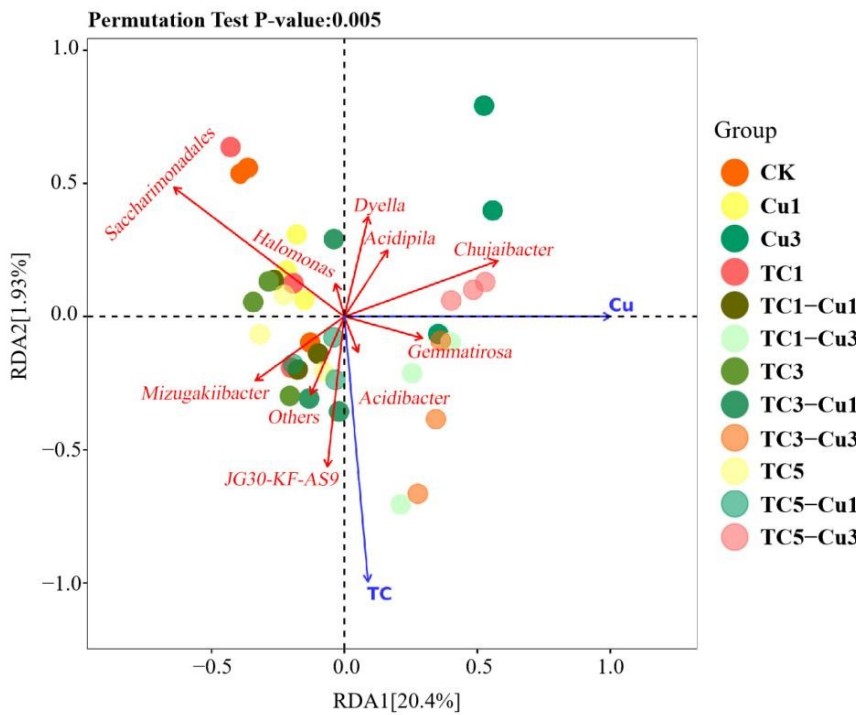

**Figure 7.** RDA ordination plot showing the relationship between the bacteria community structure and the environmental factors. (Blue arrows represent environmental factors; Red arrows represent species names.)

## 4. Conclusions

In conclusion, we explored the effects of single and combined pollution by TC and Cu on the growth of water spinach, and the structure of soil microbial community were studied by pot experiment. First, the inhibition of physicochemical indicators of water spinach by single Cu treatments was more significant than TC treatments, and the toxic effects of TC and Cu on water spinach showed synergism at low concentrations and antagonism at high concentrations. In this study, Cu had the greatest effect on the stress of water spinach, followed by TC-Cu interaction, and TC had the least effect. Second, both single TC and Cu treatments reduced species richness and diversity in the soil. The results of cluster heat map analysis showed that Cu had a greater effect on the composition of the genus level microbial community than TC, and TC (either single or combined) had a negligible effect on community composition at the genus level at low concentrations, and a greater effect at high concentrations. Finally, RDA analysis showed that both TC and Cu had effects on bacterial community structure, where the genus with strong positive correlation with TC were *JG30-KF-AS9* and *Acidibacter*, *Chujaibacter* and *Gemmatirosa* were the genus with strong positive correlation with Cu.

**Supplementary Materials:** The following supporting information can be download at: https://www.mdpi.com/article/10.3390/pr10061135/s1, Table S1: The effects of single and combined addition of TC and Cu on seed root length.

**Author Contributions:** Investigation, J.T., J.W. and A.J.; supervision, A.J. and X.T.; writing—original draft, J.T., J.W. and X.Z.; writing—review and editing, M.Z., J.Z. and X.T. All authors have read and agreed to the published version of the manuscript.

**Funding:** This research was funded by the Key R & D Project of Guangdong Province (Nos. 2020B0202080002 & 2019B110207001).

**Institutional Review Board Statement:** Not applicable.

**Informed Consent Statement:** Not applicable.

**Data Availability Statement:** Not applicable.

**Conflicts of Interest:** The authors declare no conflict of interest.

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
