# Peer review of "Effects of Tetracycline and Copper on Water Spinach Growth and Soil Bacterial Community"

_processes, doi:10.3390/pr10061135_

Round 1
Reviewer 1 Report
I think that Animal manure is one of the most important biomass for growing some plants.
In addition, it should be more utilized.
Therefore, this manuscript has some interesting results.
However, there are some of changes needed for the better manuscript.
Please check the comments.
p.2, line 93
Why the authors select water spinach for this experiment?
p.3, line 107 to 117
What is the temperature in the greenhouse?
Temperature condition is one of the most important factors.
I suggest authors that
p.7, line 243-
How was the initial soil bacterial community?
If it is possible, authors have to show the results of test soil from the vegetable field in Panyu.
Reviewer 2 Report
This manuscript describes research and results on the effect of Tetracycline antibiotic and copper contamination of soil on plant growth (water spinach) and soil microbial community structure. The topic is of significance but the manuscript needs a major revision to justify publication. Specific comments:
- The experimental setup of applying Tetracycline and copper sulfite solutions to soil and estimating their effects on plant growth and the soil microbiome is correct. The justification of the research was to monitor soil contamination from livestock manure with residual tetracycline and copper. Therefore, additional experiments incorporating contaminated manure into soil and then measuring the effects of tetracycline and copper should have been performed. These would be more relevant to the real-life situation.
- Section 2.2:
- Bad English in this section; polish and rephrase.
- Why did you choose these concentrations of TC and Cu? Are they relevant to real-life cases of soil contamination by manure?
- Line 117: delete “CK”; replace with “check” or “control”.
- This section shows results, but these should be presented in the Results, not in Materials and Methods.
- No statistical analysis has been done for the results in Table 2. This table should be moved to Results.
- Table 3, concentrations are per kg of soil; say so in legends.
- Section 2.3.1: Why did you analyze these compounds? What is MDA? Mention your rationale. You partially did later in the manuscript in section 3.1.2, but you should move this here (and elaborate on the rationale).
- Section 2.4: The extraction and purification of DNA from soil samples is missing. The methodology should be added.
- The Conclusions are very brief and almost in the format of bullets for a presentation. Not fitting for a scientific publication.
- Most references are from China; the authors are also from China. Relevant references from scientists outside China should be included and presented in the manuscript where appropriate, for a more balanced support of the presented research.
Reviewer 3 Report
The results of this investigation are very interesting, atual and important for protection of soils. This article may be published. I want Authors success in further scientific work.
In addition, Discuss whether the established effect of the influence of the TS and Cu on soil microbial community composition depends on the type of soil.
Reviewer 4 Report
please see attached

Reviewer 5 Report
There are many points that must be taken to improve the research:
1- The abstract should be shortened to be within permissible limits.
2- I cannot understand why authors measure POD activity as indicator of oxidative stress. I prefer to measure glutathione peroxidase instead of POD because its results are more realistic than POD.
3- There is an error in the PCR program, as there are two types of the number of cycles used, which are 25 and 30 cycles. It is necessary to ascertain which one is correct.
4- There is a very clear error in the statistical analysis, especially the SD, inall figure. It is necessary to increase the replicate of the samples to reach five with the measurement of the standard error of the mean (SEM).
Round 2
Reviewer 2 Report
The revised manuscript is improved. I suggest the following corrections:
Line 35: Tetracycline (not TC; in full for the first time it appears in the main body of the manuscript)
Line 79: Botanical name of the plant in italics
Line 101, legend of Table 1: You mean test soils?
Line 206: ...and biomass (space between words)
Section 3.2.2: All names of phyla should be in italics
Figure 4, legend of phyla: All names of phyla should be in italics; make sure that names of organisms are in italics throughout the manuscript.
Line 386: Space before parenthesis
Reviewer 5 Report
The two most important parts were not changed, which is the statistical analysis, where it is necessary to measure the SEM, as well as the measurement of the activity of the glutathione peroxidase enzyme instead of POD.
